# Investigating Welfare Metrics for Snakes at the Saint Louis Zoo

**DOI:** 10.3390/ani12030373

**Published:** 2022-02-03

**Authors:** Lauren Augustine, Eli Baskir, Corinne P. Kozlowski, Stephen Hammack, Justin Elden, Mark D. Wanner, Ashley D. Franklin, David M. Powell

**Affiliations:** 1Saint Louis Zoo, 1 Government Drive, Saint Louis, MO 63116, USA; baskir@stlzoo.org (E.B.); kozlowski@stlzoo.org (C.P.K.); hammack@stlzoo.org (S.H.); elden@stlzoo.org (J.E.); wanner@stlzoo.org (M.D.W.); franklin@stlzoo.org (A.D.F.); dpowell@stlzoo.org (D.M.P.); 2Smithsonian National Zoological Park, 3001 Connecticut Avenue NW, Washington, DC 20008, USA

**Keywords:** herpetoculture, tongue flicking, fecal glucocorticoid metabolites, animal care, husbandry, reptiles, substrate

## Abstract

**Simple Summary:**

Behavioral and physiological measures can be used in tandem to evaluate the impacts of animal care on snake welfare in zoological institutions. Herein, we evaluated the impacts of disturbance in the form of changing the newspaper, or ground covering, in animal habitats on seven snakes. Increased paper changes resulted in increased glucocorticoid metabolite concentrations, but did not result in increased behaviors associated with arousal (tongue flick, exposure, locomotion). These results demonstrate the need to further investigate the behavioral and physiological responses of snakes to different aspects of animal care at a species level. Furthermore, baseline behavioral and physiological data are needed to make identifying deviations from normal levels useful in welfare assessments.

**Abstract:**

Modern herpetoculture has seen a rise in welfare-related habitat modifications, although ethologically-informed enclosure design and evidence-based husbandry are lacking. The diversity that exists within snakes complicates standardizing snake welfare assessment tools and evaluation techniques. Utilizing behavioral indicators in conjunction with physiological measures, such as fecal glucocorticoid metabolite concentrations, could aid in the validation of evidence-based metrics for evaluating snake welfare. We increased habitat cleaning, to identify behavioral or physiological indicators that might indicate heightened arousal in snakes as a response to the disturbance. While glucocorticoid metabolite concentrations increased significantly during a period of increased disturbance, this increase was not associated with a significant increase in tongue-flicking, a behavior previously associated with arousal in snakes. Locomotion behavior and the proportion of time spent exposed were also not affected by more frequent habitat cleaning. These results demonstrate the need to further investigate the behavioral and physiological responses of snakes to different aspects of animal care at a species and individual level. They also highlight the need to collect baseline behavioral and physiological data for animals, in order to make meaningful comparisons when evaluating changes in animal care.

## 1. Introduction:

Reptiles have specific husbandry needs, and failure to meet these needs can result in impaired health and welfare [1,2,3,4]. Snakes are often kept under human care in minimalistic habitats [5,6,7,8], and snake welfare is an increasingly studied topic (e.g., [7,8,9,10,11,12,13,14]), as husbandry can vary greatly between, and even within, zoological institutions. Efforts to improve herpetological husbandry can be challenging, due to knowledge gaps in reptile and amphibian biology and a lack of evidence-based husbandry practices [15,16,17,18]. Welfare- and evidence-based decision-making should be the cornerstones of good animal care. As animals respond to stress in several general ways (e.g., behavioral changes, alterations in the functioning of the autonomic nervous system, and both neuroendocrine and immune responses) [19], the development of standardized, evidence-based metrics for evaluating welfare should be as species specific as possible and include both behavioral and physiological components.

Behavioral assessments are an essential method for evaluating animal welfare [3], and changes in reptile behavior can be used to evaluate their well-being [20,21,22,23]. Snakes provide a unique opportunity to investigate environmental influences on behavior and physiology, as they are tightly linked to their environments through their limbless morphology and ectothermy [24,25]. There are challenges with using behavioral measures, such as the paucity of information available on wild snake behaviors to derive standards, the misinterpretation of abnormal behaviors, and the common perception that reptiles are highly adaptable to captivity [5]. Further complicating the development of standardized metrics is the diversity within the sub-order Serpentes, with over 3950 species presently recognized (Uetz, P. (editor), The Reptile Database, http://www.reptile-database.org, accessed on 14 November 2021).

Snakes in human care are commonly housed on newspaper or a similar ground covering that can easily be removed and replaced as needed for routine animal care [26]. When housed on paper, snakes respond with increased arousal in the form of an increased rate of tongue flicking when the paper is removed and the animal is placed into a clean environment [27]. Chemical communication plays an important role in snake behaviors [28], including feeding [29,30], sexual behavior [31], aggregation [32,33], and trail laying [34]. Tongue flicking is a known mechanism by which chemical cues are transferred to the vomeronasal organ in snakes [35,36], and several studies have used rate of tongue flick (RTF) to investigate the response of snakes to stimuli [37,38,39,40,41,42]. RTF was used to investigate the response of snakes to cage cleaning and found that the snakes were not responding to the increased handling, but rather the novel or clean environment and the removal of familiar odors [41,42,43,44]. Results indicate that high levels of cage sanitization may keep snakes constantly alert [27], a negative consequence when considering their energetic requirements as sedentary ambush predators. Given that snakes are heterothermic and eat infrequently, they must conserve energy, and so staying alert could impose an energetic cost.

Locomotion and increased tongue flicking can be considered negatively-valenced behaviors because they may reflect a considerable energetic cost to the animal, and could potentially indicate discomfort or stress [44]. Behavioral indicators, however, must be interpreted in the context of a species’ natural history. For example, tongue flicking behavior is an information-gathering behavior for snakes [35,36] and, thus, could be interpreted as a positively-valenced behavior, because it would reflect interest in the environment and stimulation of the ’seeking’ affective drive in the brain, which has been described for mammals [45]. Furthermore, some studies have shown a positive correlation between an increase in exploratory behavior and positive events such as breeding in reptiles [46,47]. However, breeding events are not always indicative of good welfare [17]. In the absence of providing a novel stimulus, such as food or a mate, an increase in tongue flicking behavior here would be interpreted as an unnecessary increase in energetic costs and a negatively-valanced behavior.

Corticosterone levels in the blood or feces of an animal can provide a measure of the neuroendocrine response to stress [20,48,49]. Acute stressors may stimulate short-term physiological and behavioral changes [50,51], whereas chronic stressors elicit a sustained response, disrupting normal behaviors and possibly leading to impaired welfare [50,51,52,53,54]. However, a species’ natural history and the context of these alterations must be considered before determining if they indicate disturbance, injury, or disease [3], or are otherwise indicative of distress. Snakes respond to stress by activating the hypothalamic–pituitary–adrenal (HPA-) axis, which releases corticosterone into the blood [55]. Fecal glucocorticoid metabolites (FGM) can be measured non-invasively in animal feces [49], a technique that is becoming increasingly popular as a tool for evaluating reptile welfare [48,49,56,57,58,59,60,61,62,63,64]. Many factors can influence FGM levels, such as diet and metabolic rate [51,65,66]. Responses can occur in response to aversive stimuli, as well as during beneficial behaviors that require activity, including courtship and copulation [67], and as responses to enrichment [68].

In this study, we assessed the behavioral and physiological impact of changing the newspaper ground covering on Wagner’s vipers (*Montivipera wagneri)* maintained at the Saint Louis Zoo. This species is kept in a small number of Association of Zoo and Aquarium (AZA) facilities, and few reports on husbandry and behavior have been published on *M. wagneri* [69,70] and closely related species [71]. Our goal was to determine if we could identify behavioral and physiological changes that might indicate responses to heightened disturbance. We predicted that frequent newspaper changes would correlate positively with increased FGMs. We also predicted that behaviors associated with arousal or investigatory behaviors, such as tongue flicking, exposure, and locomotion, would increase.

## 2. Materials and Methods

Five male and two female 5-year-old Wagner’s vipers (Table 1*)*, were housed individually in 24 cm × 39 cm × 13 cm clear acrylic drawers in a rack system. Each habitat was furnished with newspaper as ground cover, a black plastic hide box positioned towards the back of the drawer, and a water bowl in the front. Heat tape with a thermostat (KE2 Therm Solutions, Inc. 12 Chamber Drive, Washington, MO, USA) was used along the back of the drawer to provide a thermal gradient of 23.9–28.9 °C within the habitats. Only artificial ambient lighting was offered within the room. Humidity measurements were collected once per minute from 7 HOBO loggers (Onset Computer Corporation, 470 MacArthur Blvd. Bourne, MA, USA) in drawers adjacent to each vipers’ habitat (μ RH = 73.66%, min RH = 50.53%, max RH = 91.82%). During Phase 1 (18 June to 31 July 2018), snake habitats were cleaned approximately once a week (μ = 0.99 changes/weekly) as needed, changing the newspaper ground covering when soiled. In Phase 2 (1 August to 4 September 2018), the newspaper ground covering was changed daily, regardless of soiled status. Phase 3 (5 September to 30 September 2018) was a repeat of Phase 1 with newspaper changes occurring as needed (μ = 0.89 changes/weekly). Fecal samples were collected from all seven snakes from 27 March through 10 October 2018. Because Phase 1 consisted of typical husbandry practice at the Saint Louis Zoo, a longer period of time for sample collection was used in order to increase the reliability of baseline estimates. Snakes were weighed during Phase 1 on 22 July. Snakes were fed thawed frozen mice weekly throughout the study.

Animal habitats were checked for feces and/or urates between 09:00 and 09:30 each day. When a sample was present in a habitat, that animal was transferred to a holding habitat to allow for sample collection and newspaper changes. Animals were in the holding habitat for approximately 2–3 min before being transferred back into their enclosures. The entire sample was collected and stored frozen until analysis. Animals were fed weekly, between 13:30 and 14:00 on feed days.

To record behavioral observations, vipers were filmed using board security cameras (EyeCom CC-4404C, Eye Spy Electronics, St. Louis, MO, USA) with 3.6 mm color lenses mounted in plastic drawers adjacent to each subject. These cameras were attached to wood or cardboard platforms. Additional light sources were added on 31 May, 18 days prior to data collection, to the viper room to improve visibility in the lower drawers; these measures included a side-mounted bulb with a shade. Each animal was filmed from 08:00 to 17:00 daily during all three phases, for a total of 936 recorded hours per viper across 104 days, from 18 June 2018 to 30 September 2018. Five 15-min periods were reviewed for each subject for each day. The periods were approximately 08:20 to 08:35, 09:35 to 09:50, 13:05 to 13:20, 14:05 to 14:20, and 16:05 to 16:20, for a total of 75 min being sampled per snake per study day. These representative times were included so as to have both morning and afternoon observations and to avoid instances when a keeper might be actively servicing a habitat. Continuous sampling was used, and behaviors (Table 2) were scored by observers using Observer XT (Noldus Information Technology, Wageningen, The Netherlands). Inter-observer reliability was established at 80%+ via testing against three videos and comparing scores to a key.

For Phase 1, seven days were sampled from 37 observed dates, between 18 June to 24 July 2018. Only those days in which no snakes received paper changes nor feeding were considered eligible for sampling, so as to select from days with minimal disturbances that could stimulate activity. To prevent possible effects from regularly-scheduled husbandry on specific days of the week, sampled days for Phase 1 were also chosen so that each day of the week (Monday through Sunday) was represented once. If multiple days fulfilled the same criteria for selection, the final picks were made randomly; i.e., if there were multiple Tuesday observations, only one was randomly selected for analysis. Samples for Phase 2 were chosen by randomly selecting from 13 observed dates comprising the first nine and last four days of the phase (1 August to 9 August 2018 and 1 September to 4 September 2018). All snakes were disturbed daily during Phase 2, but samples were again chosen such that each day of the week was represented only once. Due to equipment failures, less observable footage was available for Phase 3. Only six sample days were selected from a possible 13 in the ranges 5 September to 7 September, 12 September to 20 September, and 23 September to 28 September 2018, again with no duplication for days of the week. These dates were selected because two cameras with the least observable footage in Phase 3 had the most visible footage on these dates; even so, one camera could only film four sample days, and another filmed only two days.

### 2.1. Fecal Hormone Extraction

Depending on the size of the sample, 0.1 g or 0.5 g of wet feces was weighed and incubated at 37 °C for 24 h in modified phosphate-saline buffer and β-Glucuronidase/Arylsulfatase (10 µL of enzyme in 0.5 mL of buffer for 0.1 g samples; 25 µL of enzyme in 2.5 mL of buffer for 0.5 g samples) (Roche Diagnostics 10-127-698001). The following day, methanol (0.5 mL for 0.1 g samples or 2.5 mL for 0.5 g samples) was added to each sample, and the samples were shaken overnight. Liquid extracts were decanted, and solids were removed through centrifugation at 4000 *g*. Supernatants were frozen at −80 °C until assay. Fecal material was placed in a drying oven overnight at 100 °C. Snake fecal samples did not contain significant amounts of hair. Thus, the concentrations of corticosterone measured should reflect the physiological state of the snake and not be influenced by the presence of rodent hair.

### 2.2. Fecal Hormone Analysis and Validation

Fecal glucocorticoid concentrations were determined using a commercially available corticosterone enzyme immunoassay (Arbor Assays DetectX^®^ Corticosterone ELISA, K014; Arbor Assays DetectX® Ann Arbor, Michigan, MI, USA). The lower and upper detection limits of the assay were 0.078 ng/mL and 10 ng/mL, respectively. Samples were diluted 1:10 with assay buffer, and assays were performed according to the manufacturer’s protocols. For all assays, standards, samples, and quality control pools were assayed in duplicate. Hormone concentrations were determined as ng/mL and then divided by the dry weight of the extracted feces to give the results as ng/g feces. Mean intra-assay variation of duplicate samples was 8.8 ± 0.7%. Mean inter-assay variation of two quality control pools was 7.7 ± 1.9%.

Fecal extracts were tested for linearity by diluting 5 samples that contained high levels of hormone by 1/2, 1/4, and 1/8 with extraction buffer. Serial dilutions measured an average of 103.2 ± 7.3% of expected values and were parallel to the standard curve (test of equal slopes, *p* > 0.10). The accuracy of the assays was assessed by adding a known amount of hormone to 5 fecal extracts that contained low values of hormone. Addition of known amounts of hormone at 2 dosage levels resulted in recovery of 91.6 ± 2.0% of the expected values.

### 2.3. Statistical Analysis:

FGM data analyses were performed in SAS^®^ Studio 3.7 (SAS Institute Inc., Cary, NC, USA). A generalized linear mixed model was used to evaluate the effect of husbandry phase (1, 2, and 3) on FGM concentrations in feces, with husbandry phase nested within individual as a fixed factor, individual as a random blocking factor, and the number of days since the last feeding as a covariate, using heterogeneous variances among individuals. Measurements on fecal samples, where the number of days since the last feeding exceeded the number of days since the change in husbandry, were excluded from the analysis, due to their ambiguity in interpretation. The least square means of the three husbandry phases were compared using non-orthogonal contrasts, due to the incomplete factorial structure of the data set at days since last feeding = 5 (average across all samples was 5.3 ± 2.7 days). Contrasts were also used to compare the means between the two sexes within each husbandry phase.

The addition of lighting to improve data collection was evaluated using FGM. Samples collected prior to adding the lights (26 March–30 May) were not significantly different from samples after, so this suggested that the addition of lighting 18 days prior to the study should not have impacted the results.

Behavioral data analyses were performed in NCSS 2020 (NCSS LLC, Kaysville, UT, USA). RTF for each viper within each phase were determined by totaling the number of flicks observed within that phase’s sampled days and dividing by the number of hours in which the snake’s head was visible. The proportion of time in which the snakes were exposed or performing general locomotion was similarly calculated by summing total durations of these behaviors across sampled days and dividing by sampled hours. Multiple regression was used to check assumptions for repeated measure ANOVA tests that compared RTF, proportions of time spent exposed, and proportion of time observed locomoting between phases of the study, with sex of each subject and phase as fixed between factor and fixed within factor variables, respectively. To examine the correlation between behaviors and FGM values, behavioral data were first log transformed, following the formula log(x + 1). Spearman’s rank correlation coefficients were calculated between all behaviors and the hormone data without differentiation by phase, because some hormone values were not available for subjects in Phases 2 and 3 during the periods in which behavioral data were collected.

## 3. Results

Snakes had their newspaper changed an average of 4.4 times in Phase 1, 35 times in Phase 2, and an average of 2.9 times in Phase 3. Overall, there was a significant increase in FGM in Phase 2 (407.9 ng/g on average, t_86_ = 10.27, *p* < 0.0001), when the paper was being changed daily, compared to Phase 1 (127.6 ng/g on average). When daily paper changes ceased in Phase 3, FGM concentrations were significantly lower compared to Phase 2 (256.9 ng/g on average, t_86_ = 4.51, *p* < 0.0001), yet still significantly higher than Phase 1 (t_86_ = 5.67, *p* < 0.0001, Figure 1). Fecal glucocorticoid concentrations were significantly higher in males than females during all three phases of husbandry: 48.4 ng/g higher in Phase 1 (t_86_ = 4.19, *p* < 0.0001), 535.4 ng/g higher in Phase 2 (t_86_ = 9.91, *p* < 0.0001), and 110.8 ng/g higher in Phase 3 (t_86_ = 2.31, *p* < 0.0233). There was also a significant positive association between the number of days since the last feeding and FGM concentrations. For every additional day, fecal glucocorticoid concentrations increased by 10.8 ng/g on average (t_86_ = 4.90, *p* < 0.0001).

Observers reviewed a total of 157.4 h of video across all three phases. Snakes were most often observed exposed (>90%) but were only viewed moving, on average, less than 18% of the time. No significant statistical differences in behaviors were detected between phases (Figure 2, Figure 3, Figure 4 and Table 3) nor between male or female vipers. No behaviors were correlated with FGM concentrations (exposed R^2^_adj_ = 0.0500, ρ = 0.087; general locomotion R^2^_adj_ = 0.067, ρ = 0.093; tongue flicks R^2^_adj_ = 0.0357, ρ = −0.022), but the locomotion and tongue flick behaviors were significantly correlated with each other (R^2^_adj_ = 0.7644, ρ = 0.829, *p* < 0.0001).

## 4. Discussion

We found that vipers responded physiologically to frequent substrate changes with elevated FGMs. These elevated levels persisted after a less frequent substrate change protocol was reinstated. We did not see any changes in behaviors or correlations between FGM and behavior. Thus, support for our predictions was mixed. While high levels of corticosterone play a primary role in energy mobilization in mammals, it is unclear how high levels impact reptiles, as their energetic demands are very different [72,73]. Some species may maintain higher baseline corticosterone concentrations in predator rich environments [74,75], preparing the animal for necessary antipredator behaviors [76]. Furthermore, corticosterone levels may play a role in mediating life history trade-offs. Higher corticosterone levels favor behaviors promoting self-maintenance and survival, and lower corticosterone levels favor reproduction [77]. For these reasons, if using FGM to assess stress in snakes in human care, it is imperative to establish ranges for baseline concentrations from which to compare and evaluate welfare at the species level and to incorporate other metrics.

Glucocorticoids in the blood are metabolized by the liver and excreted into the urine or through the bile into the gut. While in the gut, glucocorticoids are heavily metabolized by bacteria, but the sterane skeletal structure is not degraded [78], making it possible to detect glucocorticoids in fecal material. After taking into account gut passage time, patterns of glucocorticoid concentrations in feces are similar to those in plasma samples [79,80]. While most studies of corticosterone levels in snakes are sampled from blood, FGM levels have been measured in snakes [49] and this is assumed to reflect an average value when compared to blood sampling [51]. This has been validated in common garter snakes, *Thamnophis sirtalis*, showing that FGM levels did reflect circulating corticosterone levels in the blood [81].

Some studies have shown that elevated corticosterone levels are associated with changes in reptile behavior [76,82,83]. In our study, there was no correlation between the behaviors observed and FGM concentrations. Similarly, Claunch et al. [54] found that elevated blood corticosterone levels did not correlate with defense behavior in Southern Pacific rattlesnakes (*Crotalus helleri)*, and Moore et al. [84] found that red-sided garter snakes (*Thamnophis sirtalis parietalis*) did not respond to acute capture stress with changes in mating behavior, despite having elevated plasma corticosterone levels. While FGM increased in our study snakes, it may not have increased to the necessary levels to affect behavior. Mating behavior in *T. sirtalis parietalis* was not suppressed until treated with levels of corticosterone of 25 µg or more [85], suggesting that there may be a threshold that activates this adaptive hormone’s function in facilitating energy availability throughout the body [19,53,82]. This potential threshold effect was also seen in aspic vipers (*Vipera aspis*) when assessing shelter availability, digestive performance, and plasma corticosterone levels [83]. It is also possible that physiological and behavioral responses to potential stressors may be decoupled [85]. Last, it is also possible that FGM may be a more sensitive predictor of an animal’s response to change than behavior alone, making it an important component of welfare evaluations.

While previous studies have found that cage cleaning was the cause of increased RTF, not the manipulation of the snakes [41,42,43,44], the increased handling in this study could have resulted in an increase in FGM. The typical management practice at the Saint Louis Zoo is to remove venomous snakes from their enclosures during cleaning and paper changes. The impact of the increased handling in Phase 2 could have contributed to the significantly higher FGM. While this handling is an important factor that requires more investigation, it is a routine part of the care of these snakes when housed on paper, making it a necessary facet of the impacts of cage cleaning from an animal management perspective.

Vipers have relatively low metabolic rates and energy requirements [86] and are characterized by ambush foraging [87], resulting in infrequent feeding on large prey [88]. While little is known about the natural history of the mountain viper complex, due to their limited and isolated distributions [89], *M. wagneri* is described as a rock-dwelling species that inhabits grassy, mountain areas at elevations of 1200–2000 m above sea level [89,90,91,92]. Shelter availability is an important aspect of a snake’s natural history and was found to have a strong influence on stress status, with consequences on major behaviors in vipers [83]. Snakes spend a considerable amount of time under shelter, and Armenian vipers, *Montivipera raddei* were generally found under rocks or under human debris during field studies [93]. Due to their predisposition for hiding and their ambush-predator hunting strategy, natural selection may not have selected for the development of more overt behavioral responses to stress, aside from possibly moving away from negative stimuli or using cover in vipers. We posit that this, in combination with restrictions on activity that may be imposed by ectothermy, could mean that most responses to stress in snakes and other reptiles might be more similar to a conservation-withdrawal approach rather than a fight-or-flight response [94,95]. Thus, while the increase in glucocorticoids may prepare a snake for a rapid fight or flight-like response in certain threatening situations, selection may have favored a more passive response, i.e., remaining still, likely relying on camouflage to avoid the negative stimuli. Based on field observations with *M. raddei*, these snakes tend to retreat upon being disturbed (M. Wanner pers. observ.). This suggests that more passive responses may be used in response to less threatening stimuli, such as handling for paper changing and new paper substrate, or that the snakes in human care are potentially habituated to human presence.

Our finding that RTF did not change with increased substrate change is contrary to previous findings that RTF increased in response to habitat cleaning and that frequent habitat cleaning may disrupt normal behavior in snakes and specifically in Viperidae [30,39]. However, only the ground covering was replaced or cleaned in this study, while the habitats themselves were not, leaving the edges of the habitats, the hide box and water bowl in each habitat, possibly retaining some familiar odors. Conant [44] was the first to document that snakes spend a considerable amount of time and energy exploring sanitized habitats and that this elevated arousal could be a sign of discomfort or stress. As such, he recommended leaving soiled objects in an otherwise clean habitat, to help reduce this response. This response was tested by Chiszar et al. [28], who found that RTF was significantly lower when a small piece of soiled paper was left in a cleaned habitat. We, therefore, suspect that we did not see an increase in RTF in our study because habitats were not sanitized, leaving scented items in the habitats. Furthermore, the snakes were held for 2–3 min in a holding container that was not disinfected between individuals. This could have also played a role in RTF and should be considered in the future.

The increase in FGM in Phase 2 did not subside immediately in Phase 3. This slow decrease could indicate that newspaper changing is a chronic stressor altering physiology over the long-term and possibly impacting welfare. Sustained high levels of glucocorticoids or chronic stress can influence reproduction, immune function, behavior, thermoregulation, and protein metabolism [96,97,98,99], and changes in these traits could affect an individual’s survival and reproductive success [100,101]. Chronic stress in reptiles can result in the loss of body mass, reproductive failure, and increased susceptibility to disease [102]. Alternatively, FGM not returning to Phase 1 levels could suggest that the snakes had adapted to a new level of demand on their physiology. Korte et al. [103] and Sterling [104] discuss the concept of allostasis or stability through change. They argue that it may not be reasonable or even adaptive for physiological parameters to return fully to a baseline condition after a sustained challenge, because the elevated parameter (FGM in this case) is an adaptive response and the physiology may be adjusting based on a prediction provided by the environment. A longer-term study in which Phase 3 is protracted and with measurement of other physiological parameters that might better capture compromised physiological function would be useful.

Interestingly, there was a significant difference in FGM between sexes, with males being more sensitive to disturbance then females. Sex-differences in fecal glucocorticoid concentrations have been reported in a variety of taxa, including sheep (*Ovis aries*) [105], domestic cats (*Felis catus*) and dogs (*Canis lupus familiaris*) [106], rats (*Rattus norwegicus*) [107], and white-throated sparrows (*Zonotrichia albicollis*) [108]. These differences, which are thought to be result from sex-differences in steroid biosynthesis or metabolism [109], are important to keep in mind when assessing the response of animals to potential stressors. This difference in snakes has not been widely explored.

Although we found that increased FGM was associated with increased levels of disturbance in these snakes, we urge caution in the design of studies using FGM to assess welfare in reptiles. Gastrointestinal passage time in reptiles makes it difficult to assess real-time stress responses, as it represents an average value pooled in the gut of the animal over a more prolonged period than is typically the case in mammals or birds [59,64]. In this study, there was a significant positive association between the number of days since the last feeding and FGM concentrations. While this association presents challenges for assessing acute stress with FGM, this non-invasive technique is still valuable when addressing chronic stress and its potential impacts on animal welfare, though it does make identification of putative stressors more challenging, unless conditions are controlled and detailed recording of environmental stimuli is done. In mammals, a post-prandial rise in glucocorticoids is an acute response that occurs shortly after eating. However, as digestion is much slower in snakes, it is possible that this difference could be due to energy metabolism related digestion. In general, longer gut-passage times are associated with decreased FGM, as more intestinal reabsorption occurs the longer the material sits in the gut [110], but little is known about these effects in snakes

An increasing number of publications have addressed snake housing [7,8,12,13,14], and many of these have utilized behavior-based evaluations. When developing standardized metrics for welfare evaluation in snakes, it is important to consider the feasibility of implementing different metrics. Counting RTF is time intensive and FGM measurements can be costly and are reported with a time delay, making a consistent implementation of these metrics unlikely in herpetoculture. We recommend improving data collection, to include some metrics that, over time, could help develop baseline levels for comparison, making deviations from normal outcomes more apparent. For example, reviewing video footage for locomotion was far less time intensive than for RTF, and our results show that locomotion was strongly correlated with RTF, making it a much more reasonable metric. We also recommend performing routine veterinary exams on snakes that include physicals, radiographs, and blood work, to develop baseline health parameters for individuals in zoological settings.

The development of animal care standards for snakes is needed to improve their welfare. Standards should be evidence-based, easily incorporated into existing care practices, and fluid as we learn more about the biology of this diverse group of animals. For example, the inclusion of novel research, such as the understudied aspects of social behavior in snakes [111], will need to be considered for inclusion in animal care standards in the future. Furthermore, an increased focus on data collection from snakes housed in human care can help amass evidence for evaluating animal welfare. For example, it is already routine for zoological institutions to record snake defecation and shed cycles. It would take minimal increased effort to develop and use a standardized method to evaluate fecal and shed quality. For example, a complete shed in one piece would be rated a 1, whereas an incomplete shed would be rated a 2, and so one, based on predefined metrics. This would aid in the detection of potential changes in animal physiology that may be indicators of impaired welfare over time.

Moving forward, it is important to consider that snakes are long-lived animals, with relatively slow metabolisms that are impacted by every facet of their environment. It can take years or even decades for the impacts of poor welfare to manifest, making identifying the exact causes of impaired health and welfare challenging. Identifying behavioral and physiological metrics to evaluate snake welfare in real time is critical to advancing snake husbandry and improving animal welfare. The development of long-term studies investigating the impacts of animal care is needed to understand the implications of husbandry and management choices for snake welfare. Furthermore, it is critical that we take a species-specific approach, as snakes exhibit a broad range of life histories and behaviors. Lastly, it is important to also consider individual differences within a species. In the case of *M. wagneri*, there was considerable variation in FGM between individuals and significant differences in FGM between sexes. Individual differences in adrenocortical function can arise from variation in body condition, disease status, age, sex, and social status [112,113,114,115]. This demonstrates the need to develop baseline behavioral, physiological, and health parameters for individual animals, in order to identify deviations that may indicate impaired welfare.

## 5. Conclusions

This study found that while *M. wagneri* did not show increased locomotion or RTF in response to increased substrate or newspaper changes, they did have a physiological response as detected in FGM. The significant increase in FGM seen in Phase 2 of this study did not subside over the following month, indicating that snakes may demonstrate a long recovery time after experiencing a stressor or adjustment to new physiological ‘norms’. Furthermore, this study demonstrates the importance of individual variation in assessment metrics, with FGM levels being extremely variable between individuals. More studies are needed to investigate the causes of stress in zoo snakes and develop metrics with predictive power to reflect positive or negative welfare.

## Figures and Tables

**Figure 1 animals-12-00373-f001:**
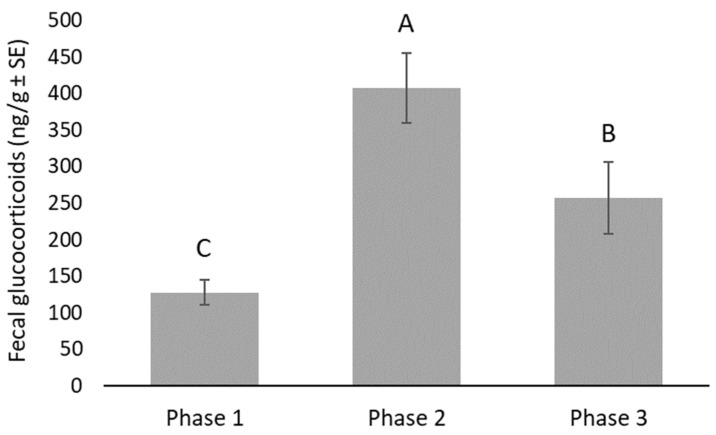
Fecal glucocorticoid concentrations (ng/g) were significantly different across the three phases of husbandry (F16,86 = 14.17, *p* < 0.0001). Pairwise differences were all declared significant at *p* < 0.05. Columns with different labels (A, B and C) are significantly different from one another.

**Figure 2 animals-12-00373-f002:**
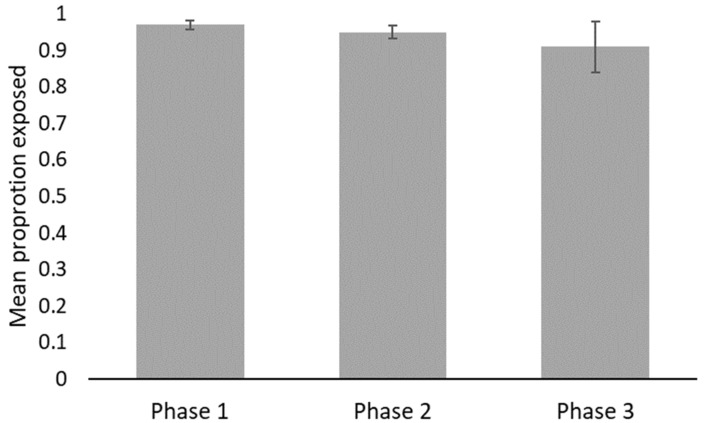
Mean proportion of time (±S.E.) vipers were observed exposed. No significant differences were detected between phases.

**Figure 3 animals-12-00373-f003:**
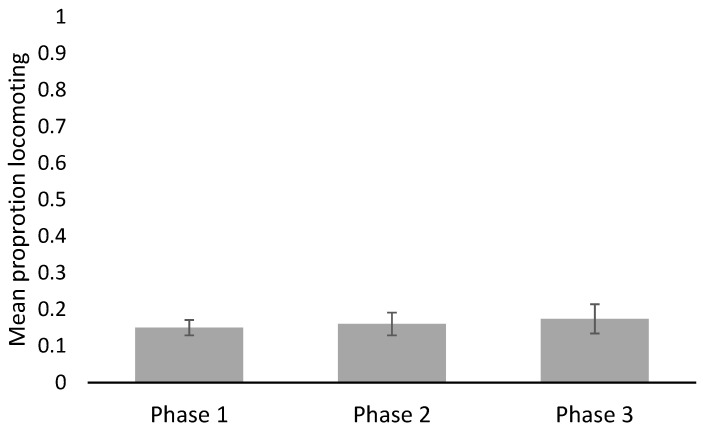
Mean proportion of time (±S.E.) vipers were observed performing general locomotion. No significant differences were detected between phases.

**Figure 4 animals-12-00373-f004:**
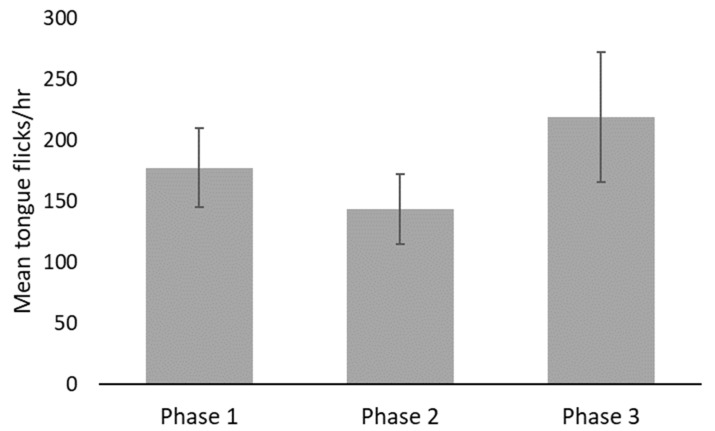
Mean rate (±S.E.) of viper tongue flicks observed per hour. No significant differences were detected between phases.

**Table 1 animals-12-00373-t001:** Seven Montivipera wagneri housed in the study, and the number of fecal samples used for FGM assessment.

Accession Number	Sex	Weight (g) July 2018	Number of Samples Phase 1	Number of Samples Phase 2	Number of Samples Phase 3
114093	Male	106	12	3	2
114094	Male	113	13	0	2
114095	Male	109	14	0	1
114096	Male	112	12	1	2
114098	Female	132	12	1	1
114099	Female	161	12	1	0
114100	Male	150	12	0	3

**Table 2 animals-12-00373-t002:** Behavioral indicators of arousal.

Behavior	Description
Hiding	Entire body concealed in the hidebox. Animal’s head and up to an additional head’s length of neck may be visible. Mutually exclusive with Exposed. State.
Exposed	Body is visible outside of the shelter. Mutually exclusive with Hiding. State.
Locomotion	Active movement of any part of body (except for tongue, see below). State.
Tongue Flick	Each discrete instance of tongue exiting mouth and re-entering mouth. Event.
Head out of sight	Snake’s head is not visible, such that the front of its mouth is unseen. State.

**Table 3 animals-12-00373-t003:** Phase means, minimum, and maximum values of exposed, locomotion, and tongue flick behaviors observed in vipers using continuous sampling.

Behavior	Phase	Min	Max	Mean	SE
Exposed (proportion of observations)	1	0.91	1.00	0.97	0.013
	2	0.88	1.00	0.95	0.017
	3	0.50	1.00	0.91	0.069
General locomotion (proportion of observations)	1	0.06	0.21	0.15	0.021
	2	0.08	0.30	0.16	0.031
	3	0.06	0.39	0.174	0.040
Tongue flick (occurrences per hour)	1	51.79	275.50	177.50	32.351
	2	51.59	242.65	143.65	28.519
	3	102.21	517.43	219.07	53.217

## Data Availability

The data presented in this study are available on request from the corresponding author.

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
