# Peer review of "Investigating Welfare Metrics for Snakes at the Saint Louis Zoo"

_animals, 2022, doi:10.3390/ani12030373_

Round 1
Reviewer 1 Report
I found this manuscript to be a nice read on a subject that is gaining importance in the fields of both snake biology as well as captive husbandry. I have a few general comments, and some specific concerns.
General Comments
- I don’t think the big range of tongue flick rate is helpful, as the rate can be adjusted pretty quickly in response to different cues. Could the rates be broken down into finer disturbance levels? Some of the work by Chiszar (cited in the manuscript) and William Cooper would support this. Tongue flicking as the authors note is often increased right after substrate changes, but is unlikely that the increased rate would be sustained during the hours-long timeline in these video observations.
- Do we know, in wild or captive snakes, what long-term stress does to species in general?
Specific Comments
- Line 75 – Could you expand on this idea more? How does being on high alert relate to the energetic requirements of snakes?
- Lines 77-86 – The authors flip-flop on whether tongue flicking is indicative of a positively-valenced or negatively-valenced behavior. For the purposes of this paper, which of the true is more likely, and how does it relate to the data/conclusions you found?
- Line 128 – Snakes are infrequent feeders whose digestive tracts are upregulated during feeding events. Do the authors think that feeding the vipers once every week may also impact their behavioral observations? Glaudas and Alexander 2017 found that frequent feedings impacted activity levels in wild puff adders, altering their spatial ecology.
- Line 136 – How long did vipers have to acclimate to the increased lighting? That may have also impacted stress and behavior.
- Figures – All four figures seem to have different “fonts” for the error bars
- Figure 1 – Vertical axis text overlaps the numbers, but may be the PDF version I am reading
- Table 4 – I am not sure about the necessity of Table 4. This may be better suited to examples within the discussion as opposed to a standalone table.
- Lines 282-298 – The authors spend this whole paragraph detailing how their results compare to other work on corticosterone measures in snakes. However, all of the cited studies relied on blood measurements of corticosterone while the authors are measuring fecal levels. While comparable, the authors should put (either here or in the introduction) a few sentences that clarify how blood and fecal corticosterone levels relate to each other.
- Line 365 – Is it possible the elevated FGM near feeding days was the result of energy mobilization?
Author Response
Hello and thank you for the feedback. Responses attached.

Reviewer 2 Report
Establishing metrics for the evaluation of husbandry practices and associated stress is important, particularly for those species which do not demonstrate easily observed, overt behaviors. So I feel this an important study. However, I had some questions about the study that I feel could be addressed to improve clarity and possible conclusions:
1.) Have FGM levels actually been validated for this species or similar species? If a FGM is elevated in a fecal sample, what is the temporal correlation with elevated corticosterone in the blood? The FGMs remained elevated in Phase 3, so does that indicate that there is a significant lag time or persistence between a stressor and elevated FGMs? Feeding and gastrointestinal passage time are factors, so does that complicate interpretation of your results with respect to cause and effect (you do address this somewhat in the discussion)? The frequency or variability of food uptake, defecation, and the number of samples per individual per phase seems to be important information that is not presented.
2) Because newspaper changes were important to the study it would be important to know more details about the process. Were individuals placed in a communal holding habitat (as per fecal collection) and if so, for how long? Was there a suitable thermal gradient in the holding habitat and was it cleaned between individuals? Would a communal holding habitat with all its associated odors, etc. be the actual stressor rather than the handling?
3) There seem to a be a couple of husbandry details that are lacking. A better description of the hide box would be helpful. Was the photoperiod artificial or natural?
4) A bit more information about the natural history of the Wagner's Viper would be helpful. Is there a seasonality to their activity? Is there a discrete breeding season? Do they go through brumation? These might also affect FGM levels. Is the timing of study (mid June to late September) appropriate for the life history (i.e. are there any seasonal factors that may affect your results)?
5) The FGM levels were significantly elevated in males compared to females, but this is not really addressed as to possible reasons. It does seem to be important for management.
6) Is Table 4 appropriate for this presentation or should suggested additions to data collection be dealt with in the discussion?
Author Response
Thank you for your feedback. I have addressed your questions in the attached document.
